# Identification, Characterization, and Regulatory Mechanisms of a Novel EGR1 Splicing Isoform

**DOI:** 10.3390/ijms20071548

**Published:** 2019-03-28

**Authors:** Vincenza Aliperti, Giulia Sgueglia, Francesco Aniello, Emilia Vitale, Laura Fucci, Aldo Donizetti

**Affiliations:** 1Department of Biology, University of Naples Federico II, 80126 Naples, Italy; vincenza.aliperti@unina.it (V.A.); giulia.sgueglia@live.com (G.S.); faniello@unina.it (F.A.); fucci@unina.it (L.F.); 2NeurOmics Laboratory, Institute of Protein Biochemistry (IBP), CNR, 80131 Naples, Italy; e.vitale@ibp.cnr.it

**Keywords:** alternative splicing, EGR1, exitron, gene expression regulation, immediate early genes

## Abstract

EGR1 is a transcription factor expressed in many cell types that regulates genes involved in different biological processes including growth, proliferation, and apoptosis. Dysregulation of EGR1 expression has been associated with many pathological conditions such as tumors and brain diseases. Known molecular mechanisms underlying the control of EGR1 function include regulation of transcription, mRNA and protein stability, and post-translational modifications. Here we describe the identification of a splicing isoform for the human *EGR1* gene. The newly identified splicing transcript encodes a shorter protein compared to the canonical EGR1. This isoform lacks a region belonging to the N-terminal activation domain and although it is capable of entering the nucleus, it is unable to activate transcription fully relative to the canonical isoform.

## 1. Introduction 

Early growth response protein 1 (EGR1) is a zinc finger transcription factor encoded by gene mapping on the human chromosome 5 and consisting of two exons. The modular structure of the EGR1 protein consists of a strong N-terminal and a weak C-terminal activation domain, an inhibitory domain, a DNA-binding module composed of three zinc-finger domains, and a bipartite nuclear localization signal [1]. The inhibitory domain between the N-terminal activation domain and the DNA-binding domain binds two transcriptional co-factors termed NGFI-A binding proteins 1 and 2 (NAB1, NAB2) that block the biological activity of EGR1 [2,3,4]. EGR1 is a pleiotropic mediator whose corresponding gene acts as a convergence point for many signaling cascades, Raf/Mitogen-activated protein kinase/ERK kinase (MEK)/Extracellular-signal-regulated kinase (ERK), Rho/ Rho-associated protein kinase (ROCK)/ LIM kinase (LIMK), Four and a half LIM (FHL), cyclic AMP (cAMP)/protein kinase A (PKA), and protein kinase C (PKC) and, upon induction, contributes to several cellular processes including proliferation, apoptosis, and differentiation in a cell type- and stimulus-specific manner [5,6]. The role of EGR1 is particularly well documented in the adult nervous system where it acts as an activity-dependent transcription factor highly expressed during long-term potentiation (LTP) and long-term depression (LTD) and, as such, is often used as a marker of neuronal stimulation. Indeed, EGR1 affects synaptic plasticity through regulation of target genes contributing to higher order processes such as learning and memory and is considered to be involved in pathological states such as addiction, anxiety, and neuropsychiatric disorders [7,8,9,10]. Moreover, EGR1 has significant tumor suppressing properties and regulates the expression of several tumor suppressor genes such as transforming growth factor beta 1 (*TGFβ1*), phosphatase and tensin homolog (*PTEN*), and tumor protein P53 (p53) [11]. EGR1 expression is low in many tumor types, such as breast, lung, glioblastoma, astrocytoma, and fibrosarcoma [11,12,13,14,15,16,17] tumors. Deletion of the *EGR1*-containing locus 5q31 has been observed in myelodysplasia and acute myelogenous leukemia [18,19,20]. It acts as a tumor suppressor via several mechanisms including cell growth repression, inhibition of apoptosis, and enhanced cell adhesion [12,14,21,22,23,24,25]. EGR1 can act as an oncogene in other conditions and is indeed present at higher levels in all types of human prostate tumors relative to normal cells [13,26,27,28]. EGR1 overexpression is well correlated with the loss of its co-repressor NAB2 in primary prostate carcinoma; this unbalance between EGR1 and NAB2 expression results in high EGR1 transcriptional activity in prostate carcinoma cells [29]. The molecular mechanisms underlying the oncogenic effect of EGR1 in the prostate remains unknown. It is of note, however, that several genes regulated by EGR1, such as *TGFβ1* and platelet derived growth factor subunit A (*PDGFA*)*,* have been proposed as growth promoters for prostate epithelial cells [28,30]. The expression and consequently the activity of EGR1 is finely regulated, moreover, disease-associated mutations in the coding sequence of *EGR1* have not yet been described. The mechanisms underlying the regulation of its expression are necessary to unravel its involvement in normal and pathological conditions. The activity of EGR1 is under strict control through the regulation of its expression at many levels, ranging from transcription to protein stability. The *EGR1* gene belongs to the group of immediate early genes, as stimulation with sera or growth factors, including nerve growth factor (NGF) and brain-derived neurotrophic factor (BDNF), rapidly and transiently induce *EGR1* gene expression [31,32,33,34,35,36]. The transcription factor function depends widely on post-translational modifications that affect either stability or activity. In particular, EGR1 can be sumoylated by the Ubiquitin conjugating enzyme (Ubc9)/Small ubiquitin-related modifier 1 (SUMO-1)/Cyclin dependent kinase inhibitor 2A (ARF) system by a mechanism dependent on the phosphorylation by protein kinase B (AKT) [37]. The sumoylation of EGR1 may be involved in its ubiquitination and degradation, thereby affecting protein stability [38]. A complex balance between acetylation and phosphorylation was also hypothesized to affect the transcriptional activity and the stability of EGR1 underlying the regulation of cellular life-and-death responses by transactivation of various target genes by EGR1 [39]. Most commonly, regulation is achieved by phosphorylation of the transcription factor that, in turn, modulates either its transactivating or DNA-binding activity [10,37,40]. For instance, while EGR1 phosphorylation levels are significantly low in unstimulated cells, EGR1 protein production induced by growth factors undergoes substantial phosphorylation events involving protein kinase C (PKC) or tyrosine kinases; the result is an increase of its DNA-binding activity [10]. Instead, the hyperphosphorylation of EGR1 by the protein kinase, casein kinase II (CKII), has a negative effect on its DNA-binding and transcriptional activity [41]. It suggests that the phosphorylated form of EGR1 plays distinct roles in cellular physiology [42] and the phosphorylation/dephosphorylation process may serve as a molecular switch for restricting the function of EGR1 as a transcription factor [43]. In addition, both in mouse and human, EGR1 is regulated at a post-transcriptional level by alternative polyadenylation, a mechanism that generates two *EGR1* mRNA variants differing for a potential N-methyl-D-aspartate receptor (NMDA-R)-responsive Carboxypeptidase E (CPE) sequence [44,45]. No additional molecular mechanism has been characterized for EGR1 functional control beyond regulation of transcription, alternative polyadenylation, mRNA and protein stability, and post-translation modifications. Our report shows the identification and characterization of a splicing event involving the *EGR1* gene transcript that clearly removes a region in the coding sequence of the second exon, strongly affecting the EGR1 transcriptional activation property.

## 2. Results

With the aim to clone the entire coding region of the transcription factor *EGR1*, we designed PCR reactions using a pair of primers spanning from the start codon to the stop codon based on the mRNA sequence reported in the National Center for Biotechnology Information (NCBI) database (NM_001964.2). As a template, we used a cDNA sample obtained by the retrotranscription of RNA extracted from the SH-SY5Y neuronal cell line differentiated by retinoic acid (RA). The PCR reaction products were subjected to gel electrophoresis and, surprisingly, we found two different amplicons—one of the expected size and the additional one with a smaller size relative to the first (Figure 1A).

We isolated and cloned both gel bands to identify the corresponding sequences. As expected, the amplicon with the highest molecular weight corresponded to the entire coding sequence of *EGR1* reported in genome databases; however, the smaller amplicon corresponded to the *EGR1* transcript lacking a central region of 414 bases in length in the coding DNA sequence (CDS) (Figure 1B). The bioinformatic translation of the alternative EGR1 isoform, hereafter named EGR1 Δ141–278, showed that it retained the correct frame and encoded for a shorter protein compared to the canonical EGR1 (Figure 1B). The absence of this region does not alter the ability of the protein to translocate into the nucleus. In fact, we analyzed the cellular localization of the protein after transfection of expression vectors for both the canonical and alternative EGR1 isoforms in HEK293T cells. Western blotting experiments on nuclear and cytoplasmic fractions showed that the EGR1 Δ141–278 isoform preferentially displayed nuclear localization (Figure 2A), as corroborated by immunofluorescence analysis (Figure 2B).

The nuclear localization led us to hypothesize that the alternative EGR1 isoform may affect gene transcription. To evaluate this hypothesis, we analyzed the expression level variation of different EGR1 target genes. We planned iper-expression experiments for both the canonical and alternative isoforms and qPCR assays on putative target genes. Based on the list of EGR1 target genes provided by Duclot and Kabbaj [10], we chose several genes to be tested in a preliminary survey to identify suitable candidate genes capable of responding to EGR1 activity under our experimental conditions. These genes included activity regulated cytoskeleton associated protein (*ARC*), salt inducible kinase 1 (*SIK1*), cyclin-dependent kinase 5 activator 1 (*P35*), activating transcription factor 3 (ATF3), apolipoprotein E (*APOE*), BCL2 associated X apoptosis regulator (*BAX*), tumor protein P73 (*P73*), and *C6ORF176*. Based on qPCR analysis after EGR1 iper-expression experiments in HEK293T cells, we selected two genes that increased their expression level only in the presence of an EGR1 expressing vector and not in the presence of an empty vector (data not shown): *ARC* and *SIK1*. We observed the expression of both EGR1 isoforms following 24 h of transfection (Figure 3A) and analyzed the expression level variation of the two target genes.

For both genes, the canonical EGR1 isoform was able to significantly increase transcription whereas the alternative isoform lacked the ability to fully activate transcription (Figure 3B).

We explored the occurrence of the splicing event that generates the EGR1 alternative isoform in different cell lines. The expression pattern clearly indicated that EGR1 Δ141-278 invariably displayed relatively low levels, with mean Cycle threshold (Ct) and standard error of the mean (SEM) values of 28.1 ± 0.03 (SH-SY5Y), 32 ± 0,94 (HEK293T), 29.2 ± 1.1 (HT-29), 31 ± 0.74 (U2OS).

## 3. Discussion

The identification of a new splicing isoform is of particular interest considering that alternative splicing is one of the primary mechanisms for regulating transcription factor activity [46]. It is worthy to note that the splicing event here identified highly resembles the recently described splicing event belonging to a specific type of intron retention category that generates the so-called “exitrons” [47]. Exitrons are defined as introns within protein-coding exons that, when retained, maintain the protein-coding potential of the transcript and contribute to transcriptome diversity in specific situations [47,48]. In the case of EGR1 Δ141–278, the absent amino acid region belonged to the strong N-terminal activation domain of EGR1, and it corresponded to one of the two regions identified and described by Gashler and colleagues [1] as important for its activity. Interestingly, Marquez and colleagues [47] identified several human exitrons that have sizes of multiples of 3 nucleotides and among them they already reported an exitron for the human *EGR1* gene. We observed that this exitron of 84 bases in length encodes for the second zinc-finger domain of the protein, leading to the idea that exitrons might encode for a specific protein domain. The region lacking in the EGR1 Δ141–278 sequence contains different regulation elements of protein stability and protein–protein interaction. In fact, it has been shown that EGR1 can interact with yes associated protein 1 (YAP-1) through a PPxY motif within the region lacking in the new isoform. The EGR1–YAP-1 interaction is important for the regulation of BCL2 associated X apoptosis regulator (BAX) expression, playing a crucial role in EGR1-mediated apoptosis [49]. In addition, the region lacking in the EGR1 isoform contains Lysine 272 that is involved in the sumoylation process and in several phosphorylation sites, suggesting that the described isoform may possess differing stability and activity compared to the canonical EGR1. Our results on gene expression regulation by the splicing isoform are in accordance with the fact that EGR1 transcription factor activation and repression domains function as independent modules, and that the EGR1 Δ141–278 isoform lacks a region of the activation domain, yet retains the repressive domain [1,2], likely retaining a different ability to regulate the transcription compared to the canonical isoform. We explored the occurrence of the splicing event that generated the EGR1 alternative isoform during SH-SY5Y differentiation; the alternative EGR1 transcript isoform shared a similar expression pattern but its levels were significantly lower when compared to the canonical mRNA isoform (data not shown). The Ct values obtained from the qPCR analysis in different cell lines were relatively high, reflecting overall low expression levels of this isoform. This is in accordance with the observation of Marquez and colleagues [47] of the generally low level and low efficiency of the exitrons. In this regard, it is likely that other unexplored conditions are needed to increase the expression of this splicing isoform. In conclusion, in the present work, we report the identification of an alternative splicing isoform for the human *EGR1* not previously described in the literature. The splicing event responsible for generating this isoform likely belongs to the mechanism of intron retention concerning the recently described exitrons [47], corroborated by its relatively low expression level. This isoform lacks a region of the N-terminal activation domain and shows a different ability to activate transcription relative to the canonical isoform. Considering that the new EGR1 isoform also lacks other regulatory elements—likely affecting its stability and protein–protein interaction pattern—we propose that this splicing event may be highly relevant for EGR1 function and, therefore, requires precise regulation. The presently described alternative splicing event adds a further regulatory mechanism for EGR1 transcription factor activity and lays the foundation for future investigations to better characterize its expression pattern and its role in both normal and pathological conditions.

## 4. Materials and Methods

### 4.1. Cell Cultures

The SH-SY5Y (human neuroblastoma, ATCC), HEK293T (human embryonic kidney, ATCC), HT-29 (human colon cancer, ATCC), and U2OS (human bone osteosarcoma, ATCC) cell lines were grown and propagated in Dulbecco’s Modified Eagle Medium (DMEM, EuroClone, Milan, Italy) supplemented with 2 mM l-glutamine (EuroClone), and a solution of 1% penicillin/streptomycin (EuroClone) and 10% fetal bovine serum (FBS, EuroClone).

### 4.2. RNA Isolation, Retrotranscription, RT-PCR, and qPCR

Total cellular RNA was isolated using TRI Reagent^®^ (Sigma-Aldrich, Milan, Italy) according to the manufacturer’s instructions. Different cell lines were plated at a concentration of 1.25 × 10^6^ in 60 mm plates and harvested after 36 h at the stage of active proliferation and asynchronous phase. For cellular differentiation, the N-type SH-SY5Y cells were starved for 24 h by reducing FBS (1.5%) and then treated with 10 μM retinoic acid (RA, Sigma-Aldrich) for 48 h. The concentration and purity of the RNA samples were assessed using NanoDrop^®^ 1000 (Thermo Fisher, Milan, Italy). cDNA was synthesized from 1 μg of total RNA using an Invitrogen SuperScript^®^ III Reverse Transcriptase kit (Invitrogen, Milan, Italy). Real-time qPCR analysis was performed on independent biological replicates using the SYBR green method and an Applied Biosystems^®^ 7500 Real-Time PCR System. The reaction mixture contained 50 ng of cDNA template and 400 nM of each forward and reverse primer in a final volume of 15 μL. The PCR conditions included a denaturation step (95 °C for 10 min) followed by 40 cycles of amplification and quantification (95 °C for 35 s, 60 °C for 1 min). Relative gene expression levels were normalized to the reference gene *GAPDH*. It was calculated by the 2^−ΔΔCt^ method. The primers used are listed below: GAPDH: F_5′-CGGGGACTTTGGGATGTC-3′, R_5′-CGCTTTCCGTCGTGAATTTC-3′; ARC: F_5′-GAGTCCTCAAATCCGGCTGA-3′, R_5′-GCACAGCAGCAAAGACTTT-3′; SIK1: F_5′-AAGACCGAGAACCTCCTGCT-3′, R_5′-GTGGACAGAGGCTCTCCTGA-3′; EGR1 Δ141-278: F_5’- TCACCTATACTGGCCGCTTT-3’, R_5’- GCGATCACAGGACTCCACT-3’. For the EGR1 Δ141–278 isoform expression analysis, the designed pair of primers spanned the exitron sequence. The expected amplicon length was 236 bp. The specificity of the qPCR reactions was checked by agarose gel run of the amplification products. The gel electrophoresis confirmed that only one gel band of the expected size was produced in our qPCR conditions. The specificity of qPCR reactions was further confirmed by sequencing analysis of the amplification product by Sanger method performed by external service (Bio-Fab, Rome, Italy).

### 4.3. Statistical Analysis

The results from independent biological replicates in triplicate are expressed as mean ± SEM. Statistical analysis of the qPCR data was carried out using a two-tailed t-test (Prism 6 software) with a *p*-value cut-off of 0.05.

### 4.4. Western Immunoblot Analysis

Cell pellets were lysed in a Radioimmunoprecipitation assay (RIPA) buffer (50 mM Tris-HCl pH 8.8, 150 mM NaCl, 1 mM Ethylenediaminetetraacetic acid, EDTA, 0.1% Sodium lauryl sulfate, SDS, 1% Triton X-100) containing protease inhibitors (Roche, Monza, Italy), incubated on ice for 30 min, and centrifuged at 14,000 rpm for 10 min at 4 °C. The supernatant was collected and used for protein quantification by Bradford assay (Bio-Rad, Milan, Italy). For each sample, 30 μg of protein lysate was electrophoresed in SDS gel (12% acrylamide) and blotted onto a nitrocellulose membrane. The transferred membrane was blocked with 5% non-fat milk (Bio-Rad) in tris-buffered saline (TBST) buffer (100 mM Tris-HCl pH 8, 1.5 M NaCl, 0.1% Tween) for 1 h at room temperature (RT) and incubated with primary antibodies (anti-EGR1, Immunological Sciences AB-83620, 1:1000, and anti-GAPDH, Immunological Sciences MAB-91903, 1:5000) in TBST with 3% non-fat milk (Bio-Rad) overnight at 4 °C. After several washes with TBST, the membrane was incubated with the corresponding secondary antibodies (anti-rabbit, Bethtyl A120–108P, 1:5000, and anti-mouse, Bethyl A90–117P, 1:5000) in TBST with 3% non-fat milk (Bio-Rad). After several washes, immunoreactive bands were visualized using an enhanced chemiluminescence (ECL) detection kit (EuroClone) according to the manufacturer’s instructions.

### 4.5. Cloning of EGR1 Alternative Isoform and Iper-Expression Experiments

The CDS of the EGR1 alternative isoform (EGR1 Δ141–278; accession number: MK681487) was amplified by PCR using the EGR1 CDS forward and reverse primers (F_5′-GATCGGTACCATGGCCGCGGCCAAGGC-3′; R_5′-GATCTCTAGATTAGCAAATTTCAATTG-3′) containing the sequence for KpnI and XbaI restriction enzymes respectively. cDNA samples obtained by RNA extracted from N-type SH-SY5Y cells differentiated with RA (10 µM) were used as a template. RT-PCR was performed using Taq Master Mix (NEB). The reaction mixture contained 50 ng of cDNA template and 400 nM of each forward and reverse primer in a final volume of 20 μL. The PCR conditions included a denaturation step (95 °C for 2 min) followed by 38 cycles of denaturation, annealing, and elongation (95 °C for 30 s, 60 °C for 40 s, and 68 °C for 1 min). Reaction products were cloned in the pGEM-T Easy Vector (Promega, Milan, Italy) and sequenced by Sanger method performed by external service (Bio-Fab). To gain preliminary insights into the molecular mechanism of action of the alternative EGR1 isoform, we planned transient iper-expression experiments in HEK293T cells. In particular, we compared the cellular localization and the effects on the RNA transcription of both the canonical and alternative isoforms. For the canonical EGR1 iper-expression, we used the pCMV6-EGR1 vector (SC128132, OriGene, Herford, Germany). For the EGR1 Δ141-278 alternative isoform iper-expression, we prepared the expressing recombinant vector as follows. The CDS of the *EGR1* alternative isoform (*EGR1 Δ141–278*) was cloned into a pCMV3 expression vector (OriGene) by using the restriction enzymes KpnI and XbaI (Promega). We used the transfection experiment with the empty vector as a control. The transfection was carried out in 100 mm plates, wherein 4.5 × 10^6^ HEK293T cells were plated. The following day, 14 μg of DNA of each expression vector were independently transfected in the cells using Lipofectamine^®^ 3000 (Thermo Fisher) according to the manufacturer’s instructions for 24 h in a medium without FBS. After that, transfected cells were collected for western blotting, immunofluorescence and qPCR experiments. 

### 4.6. Nuclear/Cytoplasmic Fractionation

Cell pellets were lysed in fractionation buffer (20 mM HEPES pH 7.4, 10 mM KCl, 2 mM MgCl_2_, 1 mM EDTA, 1 mM EGTA, 1 mM Dithiothreitol, DTT) containing protease inhibitors (Roche), incubated on ice for 20 min, and centrifuged at 3000 rpm for 5 min at 4 °C. The pellet contained nuclei, and the supernatant contained cytoplasmic proteins. The supernatant containing cytoplasmic proteins was transferred into a fresh tube, while the pellet of nuclei was washed with fractionation buffer and centrifuged at 3000 rpm for 10 min at 4 °C. Finally, the nuclei pellet was suspended in tris-buffered saline (TBS) buffer (100 mM Tris-HCl pH 8, 1.5 M NaCl) with 0.1% SDS and sonicated (3 cycles of 30 s on and 30 s off). A Bradford assay (Bio-Rad) was used for protein quantification.

### 4.7. Immunofluorescence Analysis

HEK293T cells were grown on coverslips and transfected with the indicated expression vectors. Following 24 h of transfection, cells were fixed in 4% paraformaldehyde (PFA) for 10 min at RT, permeabilized in 0.3% PBS (phosphate-buffered saline)-Triton X-100 for 20 min at RT, and blocked in 3% PBS-BSA (bovine serum albumin) for 30 min at RT. Cells were subsequently incubated with primary antibodies (anti-EGR1, BosterBio PA2177, 1:100, and anti-Tubulin beta-III, Elabscience E-AB-20095, 1:200) for 1.5 h at RT in 3% PBS-BSA and rinsed with 1× PBS. Specific secondary antibodies conjugated with fluorophores (goat anti-rabbit CY5 Conjugate, Bethyl A120–201C5, 1:500, and sheep anti-mouse FITC Conjugate, Bethyl A90–146F, 1:500) were used for 1 h at RT in 3% PBS-BSA. Finally, the cells were washed in 1× PBS. Nuclei were stained with DAPI and coverslips were mounted using 50% glycerol in 1× PBS.

## Figures and Tables

**Figure 1 ijms-20-01548-f001:**
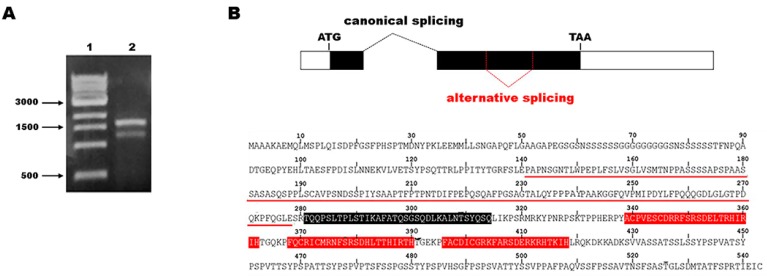
Identification of the EGR1 splicing isoform. (**A**) Electrophoresis of the PCR products related to *EGR1* coding DNA sequence (CDS) amplification. Lane 1: 1 kb marker (New England Biolabs, NEB); lane 2: PCR reaction for *EGR1* CDS. (**B**) On the top, schematic representation of the two *EGR1* transcript isoforms. The rectangles represent the two exons of the *EGR1* transcript, with the black region representing the coding sequence. Alternative splicing is indicated by dashed red lines. On the bottom, amino acid sequence of the EGR1 protein. The underlined amino acid region represents the sequence missing in the putative alternative EGR1 isoform; red sequences represent the zinc finger domains; the black sequence represents the repression domain.

**Figure 2 ijms-20-01548-f002:**
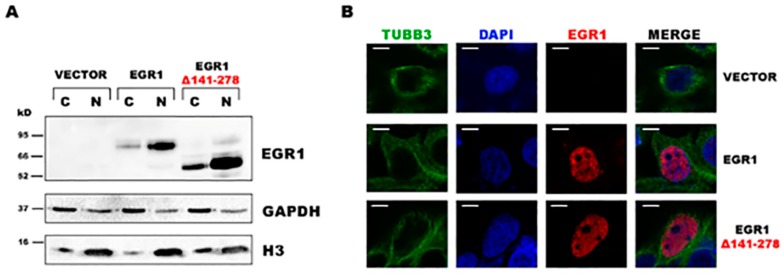
Cellular localization of EGR1 Δ141–278 splicing isoform. (**A**) Analysis of EGR1 localization by Western blotting. HEK293T cells transfected with both EGR1 constructs and an empty plasmid were subjected to cellular fractionation. Distribution of canonical and EGR1 Δ141–278 isoforms was analyzed in cytosolic (C) and nuclear (N) fractions. Glyceraldehyde-3-phosphate dehydrogenase (GADPH) and histone H3 were used as specific markers of the cytoplasmic and the nuclear fractions, respectively. (**B**) Localization of EGR1 in HEK293T cells transfected with an empty plasmid and the canonical and EGR1 Δ141–278 isoforms. Representative images of localization of EGR1 isoforms using anti-EGR1 antibody (red). The localization of the nucleus was determined by immunofluorescence with 4′,6-diamidino-2-phenylindole dihydrochloride (DAPI). Tubulin beta-III (TUBB3) immunolocalization (green) was used as a marker of cytoplasmic distribution. Scale bar = 5 µm.

**Figure 3 ijms-20-01548-f003:**
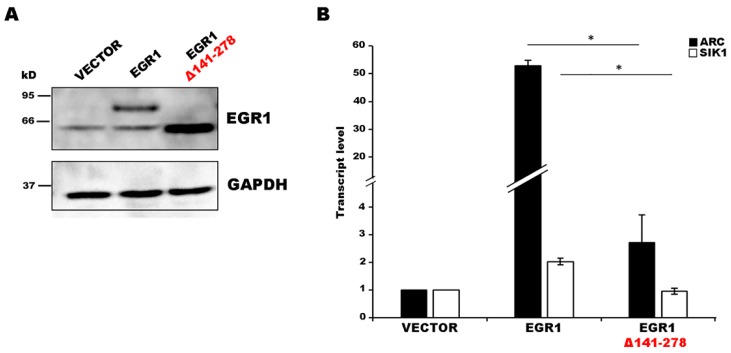
Expression pattern comparison in EGR1 and EGR1 Δ141–278 iper-expressing cells. (**A**) Analysis of the isoforms iper-expression in HEK293T cells transfected with both EGR1 constructs and an empty plasmid by Western blotting. The level of Glyceraldehyde-3-phosphate dehydrogenase (GAPDH) protein was used as a loading control. (**B**) Expression level change for activity regulated cytoskeleton associated protein (*ARC*) and salt inducible kinase 1 (*SIK1*) transcripts in EGR1 and EGR1 Δ141–278 iper-expressing cells. Gene expression level was normalized to the reference transcript *GAPDH* and calculated with the 2^−ΔΔCt^ method. The sample transfected with the empty vector was used as a calibrator. The results from independent biological replicates were expressed as the mean of fold change ± SEM (standard error of the mean). Statistical analysis of the qPCR data was carried out using a two-tailed t-test. Significance of difference in fold change (* *p* < 0.05) is shown.

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
