# Peer review of "Identification, Characterization, and Regulatory Mechanisms of a Novel EGR1 Splicing Isoform"

_ijms, 2019, doi:10.3390/ijms20071548_

Reviewer 1 Report

Aliperti et al report in their short communication the discovery of an alternative splicing variant of EGR1, that, though visualized within the nucleus, is apparently not able to activate transcription like the canonical isoform.

My comments are:

- What was the rationale behind choosing downstream genes ARC and SIK1 in this study? How did the authors perform their "preliminary survey of target genes"? Did they also look at other targets?

- Figure 1B is referenced twice

- Check for abbreviations and their explanation in the text:

    e.g. page 3, line 87: "CDS"    

- page 2 line 80: the mRNA sequence ist referenced (NM_001964.2) but it appears to be in a different format than the rest of the text

Author Response

Point 1: What was the rationale behind choosing downstream genes ARC and SIK1 in this study? How did the authors perform their "preliminary survey of target genes"? Did they also look at other targets?

Response 1: A long list of EGR1 target genes was provided by Duclot and Kabbaj (2017) based on an ENCODE dataset of ChIP experiments, and a set of these genes are under investigation in our laboratory. We wanted to analyze this set of genes in a HEK293T cell line using a qPCR approach after iper-expressing full-length EGR1. This preliminary check was done because genes transcriptionally regulated by EGR1 can be different in various cell types and under different conditions. Among the target genes analyzed were ARC, SIK1, P35, ATF3, APOE, BAX, P73, and C6ORF176. We better explain our rationale by introducing the following sentences in the text of the Results section:” We planned iper-expression experiments for both the canonical and alternative isoforms and qPCR assays on putative target genes. Based on the list of EGR1 target genes provided by Duclot and Kabbaj [10], we choose several genes to be tested in a preliminary survey to identify suitable candidate genes capable of responding to EGR1 activity under our experimental conditions. These genes included Activity regulated cytoskeleton associated protein (ARC), Salt inducible kinase 1 (SIK1), Cyclin-dependent kinase 5 activator 1 (P35), Activating transcription factor 3 (ATF3), Apolipoprotein E (APOE), BCL2 associated X apoptosis regulator (BAX), Tumor protein P73 (P73), and C6ORF176. Based on qPCR analysis after EGR1 iper-expression experiments in HEK293T cells, we selected two genes that increased their expression level only in the presence of EGR1 expressing vector and not in the presence of an empty vector (data not shown): ARC and SIK1.” 

Point 2: Figure 1B is referenced twice 

Response 2: The correction has been done. 

Point 3: Check for abbreviations and their explanation in the text: e.g. page 3, line 87: "CDS" 

Response 3: We have checked all abbreviations and any missing explanations have been included in the text. 

Point 4: page 2 line 80: the mRNA sequence ist referenced (NM_001964.2) but it appears to be in a different format than the rest of the text 

Response 4: The correction has been done

Reviewer 2 Report

 In their article, the Authors identified a new splicing isoform of EGRF1 gene, which lacks a region of the N-terminal activation domain, preferentially displayed nuclear localization and lost its ability to fully activate transcription of ARC and SIK1. EGRF1 is a transcription factor, which plays a key role in many signalling pathways such as Ras/MEK/ERK or cAMP/PKA and can act as a tumour or oncogene in many tumours. Therefore a better understanding of the regulation of EGRF1 expression is crucial, so the topic of the manuscript is interesting and highly important. The study is well planned and data are convincing, therefore this paper should be available to the scientific community however, the following points need to be addressed:

•    Why the Authors analyzed alternative splicing of EGR1 in SH-SY5Y cells? Is the new splicing isoform EGR1 Δ141-278 characterized only for these cells or could be present also in other cell lines?

•    Could the Authors explain why they chose ARC and SIK1 as EGR1 dependent? There are many other genes regulated by this transcription factor.

Author Response

Response to Reviewer 2 Comments 

Point 1: Why the Authors analyzed alternative splicing of EGR1 in SH-SY5Y cells? 

Response 1: SH-SY5Y is the main cell line that we currently use as a model of neuronal differentiation in our laboratory. To analyze the role of EGR1 in neuronal differentiation, we identified the alternative isoform. 

Point 2: Is the new splicing isoform EGR1 Δ141-278 characterized only for these cells or could be present also in other cell lines? 

Response 2: We performed a preliminary investigation on the splicing event in other cell lines and noted a relatively low expression level of the EGR1 Δ141-278 isoform, as demonstrated for the SH-SY5Y. These results are in accordance with the analysis of Marquez et al. (2015), showing that this splicing event has low efficiency. Indeed, we reported in the discussion section the following sentence, “In this regard, it is likely that other unexplored conditions are needed to increase the expression of this splicing isoform”. Essentially, the main aim of our paper was to provide a preliminary molecular characterization of the EGR1 Δ141-278 and its different impact on transcription compared to full-length EGR1; this is the main reason for not reporting data in other cell lines.

Point 3: Could the Authors explain why they chose ARC and SIK1 as EGR1 dependent? There are many other genes regulated by this transcription factor.

Response 3: A long list of EGR1 target genes was provided by Duclot and Kabbaj (2017) based on an ENCODE dataset of ChIP experiments, and a set of these genes are under investigation in our laboratory. We wanted to analyze this set of genes in a HEK293T cell line using a qPCR approach after iper-expressing full-length EGR1. This preliminary check was done considering that genes transcriptionally regulated by EGR1 can be different in various cell types and or conditions. Among the target genes analyzed were ARC, SIK1, P35, ATF3, APOE, BAX, P73, and C6ORF176. We better explain our rationale by introducing the following sentences in the text of the Results section:” We planned iper-expression experiments for both the canonical and alternative isoforms and qPCR assays on putative target genes. Based on the list of EGR1 target genes provided by Duclot and Kabbaj [10], we choose several genes to be tested in a preliminary survey to identify suitable candidate genes capable of responding to EGR1 activity under our experimental conditions. These genes included Activity regulated cytoskeleton associated protein (ARC), Salt inducible kinase 1 (SIK1), Cyclin-dependent kinase 5 activator 1 (P35), Activating transcription factor 3 (ATF3), Apolipoprotein E (APOE), BCL2 associated X apoptosis regulator (BAX), Tumor protein P73 (P73), and C6ORF176. Based on qPCR analysis after EGR1 iper-expression experiments in HEK293T cells, we selected two genes that increased their expression level only in the presence of EGR1 expressing vector and not in the presence of an empty vector (data not shown): ARC and SIK1.”

Reviewer 3 Report

The paper describes the identification of splicing isoform of EGR1.
General comments:
Although the findings are interesting and of importance, the authors fail to mention that two EGR1 mRNA variants were detected in astrocytomas, one that contains N-methyl-D-aspartate-receptor (NMDA-R)-responsive element (https://www.ncbi.nlm.nih.gov/pubmed/18489490). These findings suggest that EGR‐1 could be regulated in a sequence‐dependent manner in human astrocytomas. It would be interesting in the context of previous reports to validate the presence/localisation etc. of the newly identified splicing events  in neuronal cells. It seems likely that the mentioned changes and occurrence of various transcripts on mRNA level described in the Mittelbroon et all, 2009 paper  or the isoforms reported here divergent on the protein level may be even associated with many diseases and possibly triggered by the onset of these diseases etc .
Unfortunately, the authors choose HEK293T cells as a model system to determine the localisation patterns etc. of different splicing variants. While this is an easy system to test with it only provides limited information regarding complex EGR1 interactions that are associated with many different cells/tissues etc. Therefore, in my view, the approach seems to be a bit oversimplified to provide reliable evidence regarding  differences between 2 splicing variants isolated from a significantly different cell type than used in described experiments. I believe the paper would benefit greatly from considering testing these splicing variants in wider spectrum of cells including neuronal cultures.
Specific comments:
Lines 103-110; Figure 2.
WB Figure 2 shows higher level of expression of delta141-278 EGR1 compared to EGR1 (especially in nuclear fraction) which  has been normalised against housekeeping genes.
It seems to be contradictory to further statement in lines 159-161:
"- 159 We explored the occurrence of the 

160  splicing event that generates the EGR1 alternative isoform during SH‐SY5Y differentiation; the 

161  alternative EGR1 isoform shared a similar expression pattern but its levels were significantly lower 

162  when compared to the canonical isoform (data not shown)". 

Lines 160-162: it would be interesting to actually see these results: how big is the difference? Over which time the differentiation was monitored? Were mRNA levels used to determine these differences or WB? WB would be helpful as the mRNA stability etc. may not reflect the protein level.
Author Response

Response to Reviewer 3 Comments 

Point 1: Although the findings are interesting and of importance, the authors fail to mention that two EGR1 mRNA variants were detected in astrocytomas, one that contains N-methyl-D-aspartate-receptor (NMDA-R)-responsive element (https://www.ncbi.nlm.nih.gov/pubmed/18489490). These findings suggest that EGR‐1 could be regulated in a sequence‐dependent manner in human astrocytomas. 

Response 1: We agree with the referee on the importance of the polyadenylation mechanism in the regulation of EGR1 function. In this regard, we included the following sentences in the text of the introduction section: “In addition, both in mouse and human, EGR1 is regulated at a post-transcriptional level by alternative polyadenylation, a mechanism that generates two EGR1 mRNA variants differing for a potential NMDA-R-responsive CPE sequence [44,45]. No additional molecular mechanism has been characterized for EGR1 functional control beyond regulation of transcription, alternative polyadenylation, mRNA and protein stability, and post-translation modifications.” 

Point 2: It would be interesting in the context of previous reports to validate the presence/localisation etc. of the newly identified splicing events in neuronal cells. It seems likely that the mentioned changes and occurrence of various transcripts on mRNA level described in the Mittelbroon et all, 2009 paper or the isoforms reported here divergent on the protein level may be even associated with many diseases and possibly triggered by the onset of these diseases etc. 

Response 2: We agree with the referee regarding the importance of a better characterization of a plethora of features of the EGR1 Δ141-278 in neuronal cells or another cell type. However, this falls outside of the main aim of our paper that is focused on a preliminary characterization of the difference between the canonical and the alterative isoform on the transcriptional regulation, and on the identification of an exitron that encodes for a specific activation domain, in accordance with the analysis of Gashler and colleagues (1995). 

Point 3: Unfortunately, the authors choose HEK293T cells as a model system to determine the localisation patterns etc. of different splicing variants. While this is an easy system to test with it only provides limited information regarding complex EGR1 interactions that are associated with many different cells/tissues etc. Therefore, in my view, the approach seems to be a bit oversimplified to provide reliable evidence regarding differences between 2 splicing variants isolated from a significantly different cell type than used in described experiments. I believe the paper would benefit greatly from considering testing these splicing variants in wider spectrum of cells including neuronal cultures. 

Response 3: The main aim of this paper was a preliminary molecular characterization of the EGR1 Δ141-278 isoform. We identify this isoform in the differentiated SH-SY5Y where the transcript level of the isoform was relatively low. In addition, these cells are generally considered hard to transfect, therefore, we believe the HEK293T cell line was a suitable model to collect preliminary information on the alternative isoform considering that HEK293T are extensively used in transfection-based experiments. In fact, our approach was based on the use of an expression vector bearing the canonical and alternative EGR1 coding region and iper-expression experiments in order to understand whether the deletion characterizing the EGR1 Δ141-278 isoform affected the ability of the protein to translocate into the nucleus. Since the nuclear translocation ability of this isoform was not affected, we investigated the ability of this isoform to regulate transcription in comparison with the full-length isoform. This was of interest considering the analysis made by Gashler et al. (1995) who characterized a protein domain involved in the transcriptional activation that we demonstrated to be encoded by the transcript sequence removed in EGR1 Δ141-278 mRNA. From a molecular and evolutionary perspective, this is an interesting result  considering the idea that this exitron may encode for a specific protein domain. Although we performed investigations on the splicing event in other cell lines, we noted a relatively low expression level of the EGR1 Δ141-278 isoform, as demonstrated for the SH-SY5Y. These results are in accordance with the analysis of Marquez et al. (2015), who observed that this splicing event has low efficiency. This is the main reason why we made the following statement in the discussion section “In this regard, it is likely that other unexplored conditions are needed to increase the expression of this splicing isoform”. We agree with the referee on the importance of searching and analysing the splicing isoform expression in other cell lines and/or conditions, but this falls outside of the main aim of the paper. 

Point 4: lines 103-110; Figure 2. WB Figure 2 shows higher level of expression of delta141-278 EGR1 compared to EGR1 (especially in nuclear fraction) which has been normalised against housekeeping genes. It seems to be contradictory to further statement in lines 159-161: "- 159 We explored the occurrence of the 160 splicing event that generates the EGR1 alternative isoform during SH‐SY5Y differentiation; the 161 alternative EGR1 isoform shared a similar expression pattern but its levels were significantly lower 162 when compared to the canonical isoform (data not shown)". 

Response 4: We agree with referee about our confusing statement “We explored the occurrence of the splicing event that generates the EGR1 alternative isoform during SH-SY5Y differentiation; the alternative EGR1 isoform shared a similar expression pattern but its levels were significantly lower when compared to the canonical isoform (data not shown).” We modified the sentences as follows’ “We explored the occurrence of the splicing event that generates the EGR1 alternative isoform during SH-SY5Y differentiation; the alternative EGR1 transcript isoform shared a similar expression pattern but its levels were significantly lower when compared to the canonical mRNA isoform (data not shown). “, specifying that both isoforms were analyzed at transcript level. 

Point 5: Lines 160-162: it would be interesting to actually see these results: how big is the difference? Over which time the differentiation was monitored? Were mRNA levels used to determine these differences or WB? WB would be helpful as the mRNA stability etc. may not reflect the protein level 

Response 5: The comparison between the two isoforms during RA-induced differentiation was made at early (0,5h, 1h) and late (6h, 24h, 48h) time points after stimulation. We preferred not to report this expression analysis in detail since, as stated in the discussion section, the mRNA level of the splicing isoform was significantly lower when compared to the canonical mRNA isoform. The conclusion is that data on the expression of the alternative isoform under the conditions we tested, are only interesting considering their agreement with the observations of Marquez regarding the generally low level and low efficiency of the exitrons. We decided to not show these data in the article since they do not add further information that fits within the scope of the paper.

Round  2

Reviewer 3 Report

 The comments and justifications are generally satisfactory in the current form.
Nevertheless, I regret that the studies are not extended to other cell lines. While the authors claim that the studies are out of the scope of this paper, these studies would greatly increase its scientific merit.
Author Response

Response to reviewer comments

Point 1: The comments and justifications are generally satisfactory in the current form.
Nevertheless, I regret that the studies are not extended to other cell lines. While the authors claim that the studies are out of the scope of this paper, these studies would greatly increase its scientific merit. 

Response 1: We added information about the expression of the alternative isoform in other cell lines. In this regard, we included the following sentences in specific sections of the paper as indicated in brackets:

We explored the occurrence of the splicing event that generates the EGR1 alternative isoform in different cell lines. The expression pattern clearly indicated that EGR1 Δ141-278 invariably displayed relatively low levels, with mean Cycle threshold (Ct) and standard error of the mean (SEM) values of 28.1 ± 0.03 (SH-SY5Y), 32 ± 0,94 (HEK293T), 29.2 ± 1.1 (HT-29), 31 ± 0.74 (U2OS).” (Results).

“The Ct values obtained from the qPCR analysis in different cell lines were relatively high, reflecting overall low expression levels of this isoform.” (Discussion).

 In addition, we reported additional information on cell culture and qPCR in the Materials and Methods section, in accordance with the new results. 
